# Real-world study of adverse events associated with sodium zirconium cyclosilicate based on FDA adverse event reporting system and VigiAccess database

**Xiaona Jia, Lei Liu, Pan Wang***

Department of Pharmacy, Civil Aviation General Hospital, Beijing, China

* wp12202003@163.com

**Data availability statement:** All relevant data for this study are publicly available from the

## Abstract

The aim of this study was to investigate frequencies, types, and signals of adverse drug events (ADEs) associated with sodium zirconium cyclosilicate (SZC) used for the treatment of hyperkalemia, in order to inform clinicians of possible safety concerns linked with SZC in real-life usage. ADE reports associated with SZC were collected from both the FAERS and VigiAccess databases. Data extraction from FAERS was performed using OpenVigil 2.1, covering reports from the first quarter of 2004 through the third quarter of 2024. The VigiAccess database was retrieved for reports up to February 5, 2025. The ADEs were standardized and classified by using the preferred term (PT) and the system organ class (SOC) of the Medical Dictionary for Regulatory Activities (MedDRA) (Version 27.0). The reporting odds ratio (ROR) method and the proportional reporting ratio (PRR) method were used to screen positive signals and analyze the characteristics of ADE signals. In this study, 1384 and 1518 ADE reports related to SZC were obtained from the FAERS database and the VigiAccess database, respectively. At the SOC level, the ADEs retrieved in the two databases involved 26 SOCs, and the top 3 SOCs in terms of the number of reported cases were general disorders and administration site conditions, gastrointestinal disorders, and investigations. At the PT level, among the top 30 PTs in terms of the number of reported cases in the two databases, death, cardiac failure, weight increased, blood pressure increased, cardiac failure congestive, cerebrovascular accident, myocardial infarction, pneumonia, dizziness, dysphagia, and dyspnoea were the ADEs with higher number of reported cases not included in the drug instructions. A total of 41 positive signals were obtained after signal screening in FAERS database. Among them, the top 3 PTs in terms of signal strength were blood potassium abnormal (ROR = 180.224[119.925, 270.842]), blood potassium increased (ROR = 98.789[78.835, 123.792]), blood sodium increased (ROR = 35.248[14.624, 84.961]). Signals of cardiac disorders such as cardiac failure chronic, cardiac failure and cardiac failure congestive, signals of gastrointestinal disorders such as ileus and intestinal perforation, and signals of blood sodium increased and hypernatraemia are positive signals that deserve special

figshare repository (https://doi.org/10.6084/m9.figshare.30208981).

**Funding:** The author(s) received no specific funding for this work.

**Competing interests:** The authors have declared that no competing interests exist.

attention. In this study, the common ADEs associated with SZC were confirmed, and several intriguing novel signals not included in the drug instructions were discovered, which would provide more safety reference data for the clinical use of SZC.

## Introduction

Hyperkalemia is a condition in which serum potassium ions (K+) exceed 5.0 mmol/L. It is a common electrolyte disorder, particularly among specific patient populations such as patients with chronic kidney disease (CKD), diabetes, or heart failure, and those receiving renin-angiotensin-aldosterone system inhibitors (RAASis) [1,2]. This disease can lead to life-threatening conditions such as severe cardiac arrhythmias and sudden death [3–6]. Previous pharmaceutical interventions for the treatment of hyperkalemia were mainly organic polymer resins such as sodium polystyrene sulfonate (SPS), but it showed no selectivity for K+, resulting in poor tolerance and/or ineffectiveness [7]. Sodium zirconium cyclosilicate (SZC) is a novel oral potassium-lowering pharmaceutical agent. It is an inorganic, insoluble, and highly selective K+ binding agent that exchanges sodium and hydrogen ions for K+ or ammonium ions in the gastrointestinal tract, thereby increasing fecal potassium excretion and reducing serum potassium levels [8]. SZC demonstrated significantly greater adsorption capacity for K+ compared to other ions. And it has no impact on the utilisation of RAASis drugs. Additionally, SZC demonstrated a low propensity to induce electrolyte disorders in the body and showed no significant effect on liver function, nutritional status and other indicators, showing good tolerability and safety [4,9,10]. The SZC was first approved for the treatment of hyperkalemia in adults in the European Union and the United States in 2018, and then marketed in China at the end of 2019, and is now widely recommended and used [11,12]. Although some adverse reactions of SZC have been reported in previous clinical trial studies [13–16], there is still a lack of research on adverse reactions based on real-world data after being widely used on the market. Clinical trial studies limited by sample size and follow-up time may underestimate the incidence of infrequent or severe adverse drug events (ADEs). The VigiAccess database is a database used by the World Health Organization (WHO) to collect global adverse drug events. The Food and Drug Administration Adverse Event Reporting System (FAERS) database of the United States contains adverse event data for a series of drugs marketed in the United States [17–19]. In this study, we mined and analyzed the adverse events related to SZC obtained from the FAERS and VigiAccess databases, with a view to providing more reference information for clinical safe drug use.

## Materials and methods

### Data source

Data for this study were obtained from anonymized adverse event reports in the publicly available FAERS and VigiAccess databases. During the data collection and analysis process, all authors did not have access to information that could identify individual participants. In this study, OpenVigil 2.1 was used to collect ADE reporting data in the FAERS database. OpenVigil 2.1 (https://openvigil.sourceforge.net/) [20,21] is a pharmacovigilance platform for extracting FAERS-related data, which has been widely used and verified [22–24]. Using the generic name of "Sodium Zirconium Cyclosilicate" and the trade name of "Lokelma" as search terms, we obtained the ADE report of sodium zirconium cyclosilicate as primary suspect (PS) drug. The retrieval time is from the first quarter of 2004 to the third quarter of

2024. In addition, the VigiAccess database was also searched up to February 5, 2025 using the generic name of the drug.

## Data processing and standardization

ADE report information was standardized and classified by using the Medical Dictionary for Regulatory Activities (MedDRA) (version 27.0). Each ADE record will be assigned a preferred term (PT) and further classified into different systems according to the system organ class (SOC).

## Statistical analysis

The disproportionality analysis is a signal detection method based on a two-by-two contingency table (Table 1) with high sensitivity and is widely used for monitoring and signal mining of adverse drug reactions [25,26]. This study used the reporting odds ratio (ROR) method and the proportional reporting ratio (PRR) method in the disproportionality analysis to identify risk signals in ADE reports collected from the FAERS database. The formulas used for PRR and ROR calculation are shown in Table 2. ROR signals were defined as positive when the number of cases was $\geq 3$ and the lower limit of the 95% confidence interval (CI) was $> 1$. PRR signals were defined as positive when the number of cases was $\geq 3$, PRR $\geq 2$ and $X^2 \geq 4$ [27–31]. In this study, ADE which meets both PRR and ROR criteria is considered as a positive signal. Statistical analysis and data visualization were performed using Microsoft Office Excel 2019, OmicShare Tools [32] and R software (v.4.4.2).

## Results

### Basic information reported by ADEs

A total of 1,384 ADE reports using SZC as the primary suspect in the FAERS database were collected via the OpenVigil 2.1 pharmacovigilance platform, and 1,518 ADE reports using SZC were retrieved from the VigiAccess database. There were significantly more males than females in both databases (FAERS: 48.19% vs 27.46%, p < 0.001; VigiAccess: 52.50% vs 30.83%, p < 0.001). In terms of patients' age, patients with the age $\geq 75$ years old were the

**Table 1.** 2×2 contingency table for disproportionality analysis.

|  | Drug(s) of interest | All other drugs | Total |
|---|---|---|---|
| Adverse event(s) of interest | a | b | a + b |
| All other adverse events | c | d | c + d |
| Total | a + c | b + d | a + b + c + d |

"a" represents the number of reports with adverse events of interest of the drugs of interest, "b" represents the number of reports with adverse events of interest of all other drugs, "c" represents the number of reports with all other adverse events of the drugs of interest, "d" represents the number of reports with all other adverse events of all other drugs.

**Table 2.** The formulas used for PRR and ROR calculation.

| Algorithms | Equation | Criteria |
|---|---|---|
| ROR | ROR = ad/bc<br>95% CI = $e^{\ln ROR \pm 1.96 (1/a + 1/b + 1/c + 1/d) 0.5}$ | $a \geq 3$, lower limit of 95% CI>1 |
| PRR | PRR = a(c + d)/c(a + b)<br>$X^2 = [(ad - bc)^2] (a + b + c + d)/[(a + b) (c + d) (a + c) (b + d)]$ | $a \geq 3$, PRR $\geq 2$, $X^2 \geq 4$ |

most, followed by patients in the age group of 65–74 years old. In terms of geographical distribution, the majority of ADE reports originated from the Americas. The number of ADE reports associated with SZC showed a sustained upward trend over the reporting years, with a significant acceleration in the last two years. See Table 3 for basic information of the ADE reports.

## SOC-level analysis for ADE reports

In FAERS database, the total number of reported cases of ADE associated with SZC was 2246, involving 26 SOCs. And the reported cases of general disorders and administration site conditions (714, 31.79%) ranked first, followed by gastrointestinal disorders (280, 12.47%) and investigations (260, 11.58%). In VigiAccess database, the total number of reported cases of ADE associated with SZC was 2555, involving 26 SOCs. The top three SOCs by number of reported cases were the same as in the FAERS database, namely, general disorders and administration site conditions (801, 31.35%), gastrointestinal disorders (324, 12.68%), and investigations (286, 11.19%). The SOCs involved in the two databases were the same, and the details of the SOC distribution of ADEs reported for SZC are shown in Tables 4 and 5.

## PT-level analysis for ADE reports

In this study, the PTs involved in the ADEs reported for SZC were analyzed, with a focus on ADEs with high reporting frequency and high signal strength. The ADEs reported for SZC

**Table 3. Basic information of ADE reports.**

| | FAERS (n = 1384) | | VigiAccess (n = 1518) | |
|---|---|---|---|---|
| | *n* | % | *n* | % |
| Sex | | | | |
| Female | 380 | 24.46 | 468 | 30.83 |
| Male | 667 | 48.19 | 797 | 52.50 |
| Unknown | 337 | 24.35 | 253 | 16.67 |
| Age | | | | |
| <18 | 1 | 0.07 | 1 | 0.07 |
| 18-44 | 26 | 1.88 | 33 | 2.17 |
| 45-64 | 99 | 7.15 | 146 | 9.62 |
| 65-74 | 138 | 9.97 | 197 | 12.98 |
| ≥75 | 324 | 23.41 | 380 | 25.02 |
| Unknown | 796 | 57.51 | 759 | 50.00 |
| Geographical distribution | | | | |
| Africa | 1 | 0.07 | 7 | 0.46 |
| Americas | 1092 | 78.90 | 1222 | 80.50 |
| Asia | 243 | 17.56 | 169 | 11.13 |
| Europe | 48 | 3.47 | 120 | 7.91 |
| Unknown | 0 | 0 | 0 | 0 |
| Reporting year | | | | |
| 2018 | 6 | 0.43 | 0 | 0 |
| 2019 | 90 | 6.50 | 67 | 4.41 |
| 2020 | 110 | 7.95 | 115 | 7.57 |
| 2021 | 160 | 11.56 | 176 | 11.59 |
| 2022 | 264 | 19.08 | 296 | 19.50 |
| 2023 | 384 | 28.03 | 303 | 19.96 |
| 2024 | 388 | 28.03 | 14 | 0.92 |
| 2025 | 0 | 0 | 14 | 0.92 |

**Table 4. SOC distribution of ADEs reported for SZC in FAERS database.**

| SOC | PT | Case number | Percentage (%) |
|---|---|---|---|
| Gastrointestinal disorders | 65 | 280 | 12.47 |
| Investigations | 61 | 260 | 11.58 |
| Injury, poisoning and procedural complications | 54 | 175 | 7.79 |
| General disorders and administration site conditions | 48 | 714 | 31.79 |
| Nervous system disorders | 40 | 98 | 4.36 |
| Cardiac disorders | 30 | 125 | 5.57 |
| Metabolism and nutrition disorders | 29 | 113 | 5.03 |
| Infections and infestations | 26 | 67 | 2.98 |
| Renal and urinary disorders | 25 | 101 | 4.50 |
| Respiratory, thoracic and mediastinal disorders | 22 | 54 | 2.40 |
| Skin and subcutaneous tissue disorders | 22 | 49 | 2.18 |
| Psychiatric disorders | 18 | 27 | 1.20 |
| Musculoskeletal and connective tissue disorders | 17 | 40 | 1.78 |
| Neoplasms benign, malignant and unspecified (incl cysts and polyps) | 17 | 24 | 1.07 |
| Vascular disorders | 16 | 31 | 1.38 |
| Product issues | 12 | 16 | 0.71 |
| Surgical and medical procedures | 8 | 12 | 0.53 |
| Eye disorders | 7 | 17 | 0.76 |
| Hepatobiliary disorders | 6 | 8 | 0.36 |
| Ear and labyrinth disorders | 5 | 7 | 0.31 |
| Reproductive system and breast disorders | 5 | 6 | 0.27 |
| Blood and lymphatic system disorders | 4 | 7 | 0.31 |
| Immune system disorders | 3 | 7 | 0.31 |
| Social circumstances | 3 | 5 | 0.22 |
| Congenital, familial and genetic disorders | 2 | 2 | 0.09 |
| Endocrine disorders | 1 | 1 | 0.04 |

involved 546 PTs in FAERS (case number: 2246) and 594 PTs in VigiAccess (case number: 2555), respectively. Among the top 30 PTs in the number of reported cases, there were 28 coincident PTs in the two databases, as shown in Fig 1. The top 5 PTs reported in the FAERS database were death (n = 520, 23.15%), blood potassium increased (n = 81, 3.61%), constipation (n = 56, 2.49%), diarrhoea (n = 41, 1.83%), and oedema (n = 37, 1.65%), with the top 5 PTs reported in the VigiAccess database as death (n = 551, 21.57%), blood potassium increased (n = 83, 3.25%), diarrhoea (n = 61, 2.39%), hypokalemia (n = 56, 2.19%), and constipation (n = 53, 2.07%). Details of the top 30 PTs by number of reported cases are shown in Table 6.

After screening by ROR and PRR methods, a total of 41 positive signals were screened out in FAERS database, as shown in Fig 2. The top 5 PTs by signal strength were blood potassium abnormal (n = 24, ROR = 180.224), blood potassium increased (n = 81, ROR = 98.789), blood sodium increased (n = 5, ROR = 35.248), computer tomography abnormal (n = 3, ROR = 22.597), and azotaemia (n = 3, ROR = 15.068).

Subgroup analyses were conducted to explore sex-based differences in SZC-related adverse events, with comparison of positive signal PTs between male and female patients (Figs 3 and 4). In males, ADEs such as oedema, fluid overload, eye haemorrhage showed higher risk signals. In females, blood sodium increased, glomerular filtration rate decreased, ascites, and cardiac failure were identified as ADEs with higher occurrence risks.

**Table 5.** SOC distribution of ADEs reported for SZC in VigiAccess database.

| SOC | PT | Case number | Percentage (%) |
|---|---|---|---|
| Gastrointestinal disorders | 63 | 324 | 12.68 |
| Investigations | 71 | 286 | 11.19 |
| Injury, poisoning and procedural complications | 54 | 182 | 7.12 |
| General disorders and administration site conditions | 57 | 801 | 31.35 |
| Nervous system disorders | 40 | 115 | 4.50 |
| Cardiac disorders | 37 | 126 | 4.93 |
| Metabolism and nutrition disorders | 36 | 159 | 6.22 |
| Infections and infestations | 24 | 52 | 2.04 |
| Renal and urinary disorders | 27 | 114 | 4.46 |
| Respiratory, thoracic and mediastinal disorders | 20 | 53 | 2.07 |
| Skin and subcutaneous tissue disorders | 30 | 73 | 2.86 |
| Psychiatric disorders | 21 | 36 | 1.41 |
| Musculoskeletal and connective tissue disorders | 18 | 67 | 2.62 |
| Neoplasms benign, malignant and unspecified (incl cysts and polyps) | 14 | 20 | 0.78 |
| Vascular disorders | 18 | 39 | 1.53 |
| Product issues | 16 | 24 | 0.94 |
| Surgical and medical procedures | 10 | 22 | 0.86 |
| Eye disorders | 6 | 18 | 0.70 |
| Hepatobiliary disorders | 9 | 10 | 0.39 |
| Ear and labyrinth disorders | 4 | 5 | 0.20 |
| Reproductive system and breast disorders | 6 | 6 | 0.23 |
| Blood and lymphatic system disorders | 4 | 5 | 0.20 |
| Immune system disorders | 2 | 7 | 0.27 |
| Social circumstances | 4 | 8 | 0.31 |
| Congenital, familial and genetic disorders | 1 | 1 | 0.04 |
| Endocrine disorders | 2 | 2 | 0.08 |

## Discussion

Since its introduction to the market, SZC has emerged as the preferred regimen for the treatment of hyperkalemia. However, only a limited number of pre-marketing clinical trial studies have analyzed its safety profile, post-marketing pharmacovigilance studies based on real-world data remain insufficient. In this study, we analyzed the ADEs associated with SZC in real-world use by mining the data from the FAERS and VigiAccess databases, which provided more reference information for its safe clinical use.

This study found that the number of male patients with ADE reported in both databases exceeded that of female patients. This finding aligns with the results reported by Nilsson et al. [1] who conducted a study on the incidence of hyperkalemia within a large healthcare system. Their findings indicated that the incidence of hyperkalemia was lower in women. It is well known that hyperkalemia is prevalent among patients with CKD. A comprehensive review by Gilligan et al. [33] elucidated the prevalence and risk factors of hyperkalemia in patients with CKD, and pointed out that men were associated with higher serum potassium and a heightened risk of hyperkalemia in CKD. This gender difference in the prevalence of hyperkalemia may be a contributing factor to the observed sex disparities in reported ADEs.

In the FAERS and VigiAccess databases, the top 3 SOCs involved in ADEs reported for SZC were general disorders and administration site conditions, gastrointestinal disorders, and investigations, which were consistent with the common adverse drug reactions reported in the drug instructions. In addition, injury, poisoning and procedural complications, cardiac disorders, metabolism and nutrition disorders, nervous system disorders, and renal and urinary

**Table 6. Top 30 PTs by number of reported cases.**

| | FAERS | | | VigiAccess | | |
|---|---|---|---|---|---|---|
| | PT | Case number | Percentage (%) | PT | Case number | Percentage (%) |
| 1 | Death | 520 | 23.15 | Death | 551 | 21.57 |
| 2 | Blood potassium increased | 81 | 3.61 | Blood potassium increased | 83 | 3.25 |
| 3 | Constipation | 56 | 2.49 | Diarrhoea | 61 | 2.39 |
| 4 | Diarrhoea | 41 | 1.83 | Hypokalaemia | 56 | 2.19 |
| 5 | Oedema | 37 | 1.65 | Constipation | 53 | 2.07 |
| 6 | Cardiac failure | 31 | 1.38 | Oedema | 37 | 1.45 |
| 7 | Hypokalaemia | 31 | 1.38 | Nausea | 34 | 1.33 |
| 8 | Drug ineffective | 27 | 1.20 | Renal failure | 31 | 1.21 |
| 9 | Blood potassium abnormal | 24 | 1.07 | Off label use | 28 | 1.10 |
| 10 | Off label use | 24 | 1.07 | Product use issue | 27 | 1.06 |
| 11 | Renal failure | 23 | 1.02 | Hyperkalaemia | 27 | 1.06 |
| 12 | Weight increased | 23 | 1.02 | Blood potassium abnormal | 26 | 1.02 |
| 13 | Nausea | 22 | 0.98 | Drug ineffective | 24 | 0.94 |
| 14 | Blood pressure increased | 21 | 0.93 | Cerebrovascular accident | 23 | 0.90 |
| 15 | Hyperkalaemia | 21 | 0.93 | Cardiac failure congestive | 22 | 0.86 |
| 16 | Product use issue | 21 | 0.93 | Oedema peripheral | 22 | 0.86 |
| 17 | Cardiac failure congestive | 19 | 0.85 | Product dose omission issue | 22 | 0.86 |
| 18 | Cerebrovascular accident | 19 | 0.85 | Blood potassium decreased | 21 | 0.82 |
| 19 | Product dose omission issue | 19 | 0.85 | Weight increased | 21 | 0.82 |
| 20 | Abdominal discomfort | 18 | 0.80 | Dizziness | 21 | 0.82 |
| 21 | Myocardial infarction | 17 | 0.76 | Myocardial infarction | 19 | 0.74 |
| 22 | Pneumonia | 17 | 0.76 | Cardiac failure | 18 | 0.70 |
| 23 | Renal disorder | 16 | 0.71 | Rash | 18 | 0.70 |
| 24 | Vomiting | 16 | 0.71 | Abdominal discomfort | 17 | 0.67 |
| 25 | Blood potassium decreased | 15 | 0.67 | Vomiting | 17 | 0.67 |
| 26 | Dizziness | 15 | 0.67 | Peripheral swelling | 17 | 0.67 |
| 27 | Dysphagia | 14 | 0.62 | Blood pressure increased | 17 | 0.67 |
| 28 | Oedema peripheral | 14 | 0.62 | Dyspnoea | 17 | 0.67 |
| 29 | Dyspnoea | 13 | 0.58 | Intentional product misuse | 16 | 0.63 |
| 30 | Intentional product misuse | 13 | 0.58 | Renal disorder | 15 | 0.59 |

disorders were also the SOCs with the highest reported cases, and more attention should be paid to them in clinical application. In the PT-level analysis, most of the ADEs in the top 30 reported cases in the two databases were overlapping, proving the reliability of the results of this study to a certain extent. Among them, death, cardiac failure, weight increased, blood pressure increased, cardiac failure congestive, cerebrovascular accident, myocardial infarction, pneumonia, dizziness, dysphagia, and dyspnea were the ADEs with higher reporting frequency not included in the drug instructions, and they should be especially noted in clinical use.

In this study, according to the ROR method and PRR method, the ADEs in the FAERS database were screened to obtain positive signals, which involved 12 SOCs, including cardiac disorders, gastrointestinal disorders, general disorders and administration site conditions, injury, poisoning and procedural complications, investigations, metabolism and nutrition disorders, among others. Notably, signals such as cardiac failure chronic, cardiac failure, cardiac failure congestive, dysphagia, death, colon cancer, choking, dementia, cerebral hemorrhage, blood sodium increased, blood pressure increased, metabolic acidosis, and feeding disorder are not mentioned in the drug instructions, but these positive signals may be related to the patient's own disease or edema-related events caused by SZC. Cardiac disorders such as cardiac failure chronic, cardiac failure, and cardiac failure congestive are ADEs

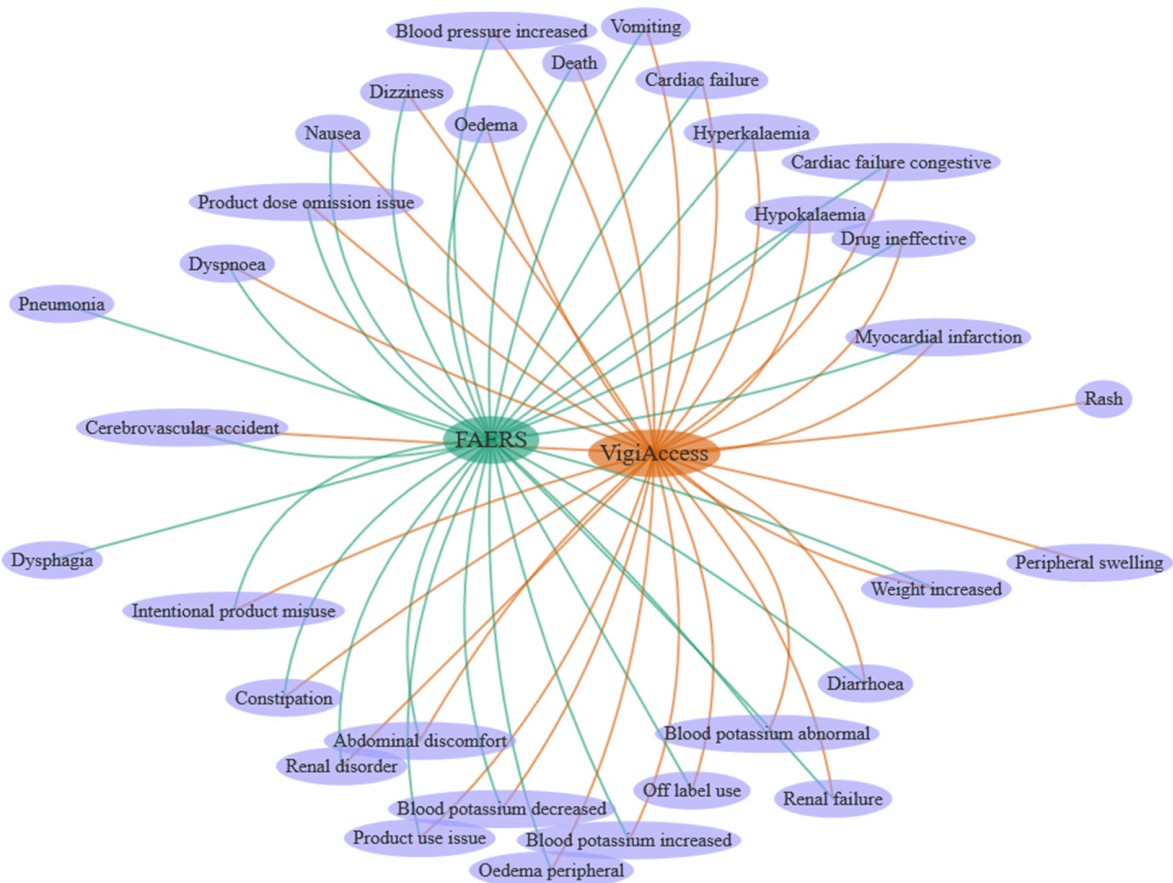

**Fig 1. Network Venn diagrams of the top 30 PTs by number of reported cases in FAERS and VigiAccess databases.**

not included in the drug instructions of SZC. In the Phase III clinical trial study of Roger et al. [34], there was a report about cardiac disorders caused by SZC, among 737 subjects, 158 subjects had serious adverse reactions, of which 10 patients had congestive cardiac failure and 4 patients had cardiac failure. The positive signals of gastrointestinal system diseases such as ileus and intestinal perforation are known severe gastrointestinal adverse events as traditional potassium-lowering drug SPS [35]. Although it is given a reminder in the precautions of SZC drug instructions, the risk situation is still unclear and more vigilance should be exercised in clinical use. Death is a positive signal of SZC, which has the highest number of reported cases. This may be due to the fact that patients with hyperkalemia are usually complicated with other diseases, such as severe heart failure and severe renal failure, and their baseline situation is poor. Multiple large-scale observational studies have shown that hyperkalemia itself is related to the increased risk of death [36–38]. Blood sodium increased and hypernatremia are two novel signals with high signal strength, which may be related to the potential sodium load existing in SZC (400mg sodium is contained in every 5g SZC). Therefore, capacity status should be monitored during use, and dietary sodium intake and diuretic amount should be adjusted in patients at risk of capacity overload [7,39]. In sex-based subgroup analyses, differences in adverse event profiles were observed. Beyond differences in signal strength for shared ADEs, sex-specific analyses revealed distinct positive signals unique

| SOC | PT | n | ROR(95% CI) | |
|---|---|---|---|---|
| Cardiac disorders | cardiac failure chronic | 3 | 12.792(4.117-39.743) | |
| | cardiac failure | 31 | 6.791(4.756-9.698) | |
| | cardiac failure congestive | 19 | 3.759(2.39-5.913) | |
| Gastrointestinal disorders | gastrointestinal motility disorder | 3 | 11.482(3.696-35.672) | |
| | ileus | 7 | 11.268(5.359-23.691) | |
| | constipation | 56 | 5.119(3.917-6.689) | |
| | intestinal perforation | 3 | 4.725(1.522-14.673) | |
| | dysphagia | 14 | 2.56(1.512-4.335) | |
| General disorders and administration site conditions | oedema | 37 | 13.008(9.381-18.038) | |
| | death | 520 | 12.782(11.456-14.262) | |
| | generalised oedema | 3 | 4.708(1.516-14.62) | |
| | oedema peripheral | 14 | 2.029(1.198-3.435) | |
| Injury, poisoning and procedural complications | incorrect route of product administration | 5 | 5.213(2.166-12.547) | |
| | product prescribing issue | 4 | 3.896(1.46-10.397) | |
| | intentional product misuse | 13 | 2.406(1.393-4.155) | |
| Investigations | blood potassium abnormal | 24 | 180.224(119.925-270.842) | |
| | blood potassium increased | 81 | 98.789(78.835-123.792) | |
| | blood sodium increased | 5 | 35.248(14.624-84.961) | |
| | computerised tomogram abnormal | 3 | 22.597(7.269-70.246) | |
| | product residue present | 5 | 14.685(6.098-35.362) | |
| | blood potassium decreased | 15 | 9.962(5.987-16.576) | |
| | glomerular filtration rate decreased | 6 | 9.445(4.234-21.069) | |
| | blood creatinine increased | 10 | 2.788(1.496-5.194) | |
| | blood pressure increased | 21 | 2.519(1.637-3.877) | |
| Metabolism and nutrition disorders | hypokalaemia | 31 | 13.774(9.645-19.671) | |
| | hypernatraemia | 3 | 12.046(3.877-37.424) | |
| | hyperkalaemia | 21 | 11.56(7.51-17.794) | |
| | fluid overload | 6 | 10.382(4.654-23.161) | |
| | metabolic acidosis | 10 | 5.124(2.75-9.546) | |
| | feeding disorder | 4 | 4.057(1.52-10.828) | |
| | fluid retention | 9 | 3.218(1.67-6.198) | |
| Neoplasms benign, malignant and unspecified (incl cysts and polyps) | colon cancer | 4 | 4.194(1.572-11.194) | |
| Nervous system disorders | dementia | 5 | 2.848(1.183-6.855) | |
| | cerebral haemorrhage | 6 | 2.768(1.241-6.173) | |
| Product issues | product taste abnormal | 5 | 5.982(2.485-14.399) | |
| Renal and urinary disorders | azotaemia | 3 | 15.068(4.849-46.82) | |
| | renal disorder | 16 | 5.427(3.314-8.885) | |
| | end stage renal disease | 6 | 4.364(1.957-9.733) | |
| | renal failure | 23 | 2.725(1.805-4.116) | |
| Respiratory, thoracic and mediastinal disorders | choking | 5 | 4.498(1.869-10.826) | |
| Social circumstances | inability to afford medication | 3 | 4.914(1.582-15.259) | |

50 100 150 200 250

**Fig 2. Positive signal detection results of SZC in FAERS database.**

to each sex. Notably, male-specific signals included eye haemorrhage and dementia, which were not detected in females. Conversely, females exhibited unique risk signals such as ascites, pulmonary oedema, and pneumonia absent in males. Although direct mechanistic evidence remains limited and requires further validation, these sex-divergent risk signals provide novel insights for pharmacovigilance and adverse reaction monitoring.

In our signal detection analysis based on the FAERS database, several positive signals were consistent with prior findings by Yu et al. [40], including cardiac failure congestive, cardiac failure chronic, ileus, intestinal perforation, blood sodium increased, hypernatremia,

| SOC | PT | n | ROR(95% CI) | |
|---|---|---|---|---|
| Cardiac disorders | cardiac failure | 11 | 3.843(2.117-6.975) | |
| | cardiac failure congestive | 11 | 3.938(2.169-7.147) | |
| Eye disorders | eye haemorrhage | 3 | 8.497(2.731-26.434) | |
| Gastrointestinal disorders | constipation | 28 | 5.68(3.889-8.296) | |
| General disorders and administration site conditions | death | 278 | 11.459(9.812-13.383) | |
| | oedema | 21 | 15.62(10.108-24.137) | |
| | oedema peripheral | 9 | 2.748(1.423-5.307) | |
| Injury, poisoning and procedural complications | intentional product misuse | 9 | 3.444(1.783-6.65) | |
| | product prescribing issue | 4 | 7.976(2.983-21.329) | |
| | incorrect route of product administration | 3 | 6.605(2.124-20.546) | |
| Investigations | blood potassium increased | 34 | 62.26(44.021-88.057) | |
| | blood pressure increased | 11 | 2.762(1.522-5.013) | |
| | weight increased | 11 | 2.155(1.187-3.912) | |
| | blood creatinine increased | 8 | 3.216(1.601-6.459) | |
| | blood potassium abnormal | 7 | 98.688(46.614-208.937) | |
| | blood potassium decreased | 3 | 5.438(1.749-16.913) | |
| Metabolism and nutrition disorders | hypokalaemia | 13 | 12.227(7.058-21.184) | |
| | hyperkalaemia | 12 | 8.755(4.944-15.503) | |
| | metabolic acidosis | 7 | 6.229(2.957-13.123) | |
| | fluid overload | 5 | 15.253(6.321-36.807) | |
| Musculoskeletal and connective tissue disorders | joint swelling | 6 | 2.754(1.233-6.154) | |
| Nervous system disorders | dementia | 4 | 4.071(1.523-10.883) | |
| Product issues | product taste abnormal | 4 | 12.552(4.692-33.575) | |
| Renal and urinary disorders | renal failure | 13 | 2.462(1.421-4.263) | |
| | renal disorder | 10 | 5.871(3.143-10.967) | |
| | end stage renal disease | 5 | 8.501(3.525-20.505) | |

**Fig 3. Positive signal detection results of SZC in male subgroup in FAERS database.**

death. Utilizing the latest data from two databases for cross-validation, our study confirmed previously reported ADEs associated with SZC while also identified several new unreported signals of potential clinical significance, such as colon cancer, dementia, cerebral hemorrhage, choking, dysphagia, and feeding disorders. Special attention should be paid to these risks in clinical practice.

This study has several limitations. Both the FAERS database and VigiAccess database are spontaneous reporting systems for adverse events. As is well recognized, such pharmacovigilance databases are subject to biases, including underreporting, duplicate entries, inaccurate or incomplete information, and other factors that may affect the analysis. In this study, OpenVigil 2.1 platform was used to mine and analyze the data of FAERS. In OpenVigil 2.1, the cleaned data were only loaded with reports with complete case information, which sacrificed the original sample size to a certain extent but ensured high data quality [20]. Although the findings of large-scale data mining provide more reference information for safe medication use, the mining results can only show that there is a statistical correlation between drugs and the detected signals, and the exact causal relationship needs further clinical verification. Future research should be grounded in the findings of big data mining, and more high-quality clinical studies should be conducted on the focused adverse reactions to identify the occurrence of relevant adverse events in clinical use.

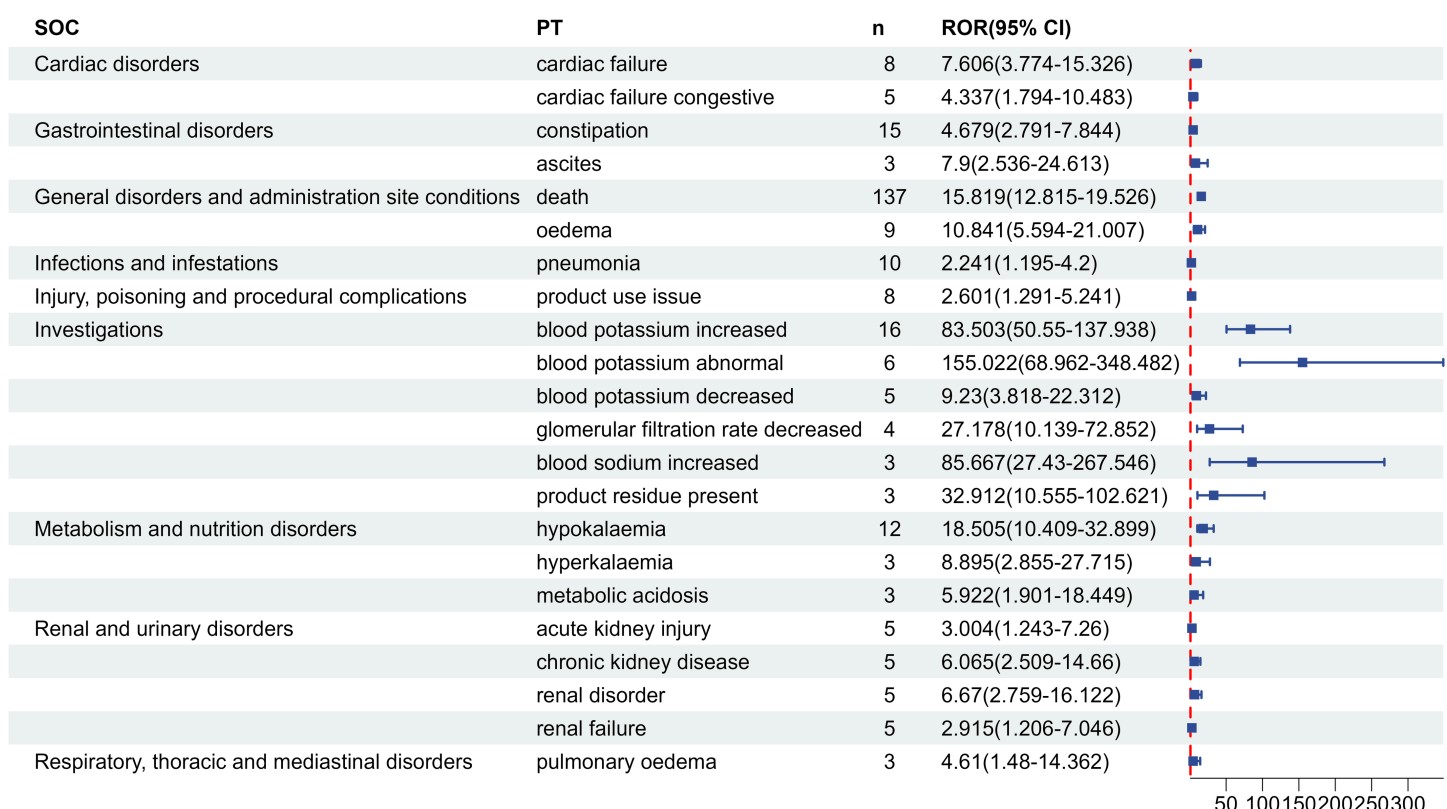

| SOC | PT | n | ROR(95% CI) |
|---|---|---|---|
| Cardiac disorders | cardiac failure | 8 | 7.606(3.774-15.326) |
|  | cardiac failure congestive | 5 | 4.337(1.794-10.483) |
| Gastrointestinal disorders | constipation | 15 | 4.679(2.791-7.844) |
|  | ascites | 3 | 7.9(2.536-24.613) |
| General disorders and administration site conditions | death | 137 | 15.819(12.815-19.526) |
|  | oedema | 9 | 10.841(5.594-21.007) |
| Infections and infestations | pneumonia | 10 | 2.241(1.195-4.2) |
| Injury, poisoning and procedural complications | product use issue | 8 | 2.601(1.291-5.241) |
| Investigations | blood potassium increased | 16 | 83.503(50.55-137.938) |
|  | blood potassium abnormal | 6 | 155.022(68.962-348.482) |
|  | blood potassium decreased | 5 | 9.23(3.818-22.312) |
|  | glomerular filtration rate decreased | 4 | 27.178(10.139-72.852) |
|  | blood sodium increased | 3 | 85.667(27.43-267.546) |
|  | product residue present | 3 | 32.912(10.555-102.621) |
| Metabolism and nutrition disorders | hypokalaemia | 12 | 18.505(10.409-32.899) |
|  | hyperkalaemia | 3 | 8.895(2.855-27.715) |
|  | metabolic acidosis | 3 | 5.922(1.901-18.449) |
| Renal and urinary disorders | acute kidney injury | 5 | 3.004(1.243-7.26) |
|  | chronic kidney disease | 5 | 6.065(2.509-14.66) |
|  | renal disorder | 5 | 6.67(2.759-16.122) |
|  | renal failure | 5 | 2.915(1.206-7.046) |
| Respiratory, thoracic and mediastinal disorders | pulmonary oedema | 3 | 4.61(1.48-14.362) |

50 100 150 200 250 300

**Fig 4. Positive signal detection results of SZC in female subgroup in FAERS database.**

## Conclusion

This study conducted a comprehensive analysis of SZC by mining real-world adverse drug event databases, systematically evaluating high frequency and high signal strength adverse events to enhance clinical safety awareness. The research not only confirmed known adverse reactions but also identified previously unreported safety signals not mentioned in the drug instructions. These findings provide clinicians with more comprehensive safety references, particularly highlighting the importance of monitoring for these newly detected adverse reactions in clinical practice. In clinical use, healthcare professionals need to pay special attention to monitoring patients' blood sodium levels, preventing cardiac risks such as chronic heart failure and gastrointestinal system risks such as ileus and intestinal perforation, as well as other risk signals detected. Pharmacovigilance research based on real-world data is imperative for providing recommendations for clinical decision-making and improving medication safety for patients.

## Acknowledgments

We sincerely thank the U.S. FDA Adverse Event Reporting System (FAERS) and WHO-VigiAccess database, which provided important pharmacovigilance data for our study.

## Author contributions

**Conceptualization:** Xiaona Jia, Pan Wang.

**Data curation:** Xiaona Jia, Lei Liu.

**Formal analysis:** Xiaona Jia, Lei Liu.

**Investigation:** Xiaona Jia.

**Methodology:** Xiaona Jia.

**Project administration:** Pan Wang.

**Resources:** Xiaona Jia.

**Software:** Xiaona Jia.

**Supervision:** Pan Wang.

**Writing – original draft:** Xiaona Jia.

**Writing – review & editing:** Xiaona Jia, Lei Liu, Pan Wang.

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
