## [Decision Letter · Decision Letter 0]

8 Jun 2025

PONE-D-25-21593Real-World Study of Adverse Events of Sodium Zirconium Cyclosilicate Based on US FDA Adverse Event Reporting System and VigiAccess DatabasePLOS ONE

Dear Dr. Jia,

Thank you for submitting your manuscript to PLOS ONE. After careful consideration, we feel that it has merit but does not fully meet PLOS ONE’s publication criteria as it currently stands. Therefore, we invite you to submit a revised version of the manuscript that addresses the points raised during the review process.

**ACADEMIC EDITOR:**
The submission reflects scientific relevance. However, some fundamental concerns have been raised by the reviewers affecting certain sections of the study. Kindly pay a thorough attention to the queries of the reviewers and address them critically before resubmission. 

We look forward to receiving your revised manuscript.

Kind regards,

Yusuf Oloruntoyin Ayipo, Ph.D

Academic Editor

PLOS ONE

Journal Requirements:

4. Please update your submission to use the PLOS LaTeX template. The template and more information on our requirements for LaTeX submissions can be found at http://journals.plos.org/plosone/s/latex.

5. Please upload a new copy of Figure 1 and 2 as the detail is not clear. Please follow the link for more information: https://blogs.plos.org/plos/2019/06/looking-good-tips-for-creating-your-plos-figures-graphics/" https://blogs.plos.org/plos/2019/06/looking-good-tips-for-creating-your-plos-figures-graphics/

6. Please remove all personal information, ensure that the data shared are in accordance with participant consent, and re-upload a fully anonymized data set.

Additional Editor Comments :

The submission reflects scientific relevance. However, some fundamental concerns have been raised by the reviewers affecting certain sections of the study. Kindly pay a thorough attention to the queries of the reviewers and address them critically before resubmission.

Reviewers' comments:

Reviewer's Responses to Questions

**Comments to the Author**

1. Is the manuscript technically sound, and do the data support the conclusions?

Reviewer #1: Yes

Reviewer #2: Partly

2. Has the statistical analysis been performed appropriately and rigorously? 

Reviewer #1: Yes

Reviewer #2: No

3. Have the authors made all data underlying the findings in their manuscript fully available?

Reviewer #1: Yes

Reviewer #2: No

4. Is the manuscript presented in an intelligible fashion and written in standard English?

Reviewer #1: Yes

Reviewer #2: No

5. Review Comments to the Author

Reviewer #1: Real-World Study of Adverse Events of Sodium Zirconium Cyclosilicate Based on US FDA Adverse Event Reporting System and VigiAccess Database

Reviewer - Charles E. Ugwu.

1. Overview

(1.I). The manuscript investigates reported ADEs from the real-world use of SZC. The descriptive statistics and disproportionality analysis used to evaluate ADEs were appropriate. The study finds novel ADE signals not included in the drug instructions. These findings will help inform stakeholders of the necessary safety considerations when prescribing or regulating this drug.

(1.II). Overall, the manuscript seeks to fill a critical knowledge gap on a clinically relevant topic but will benefit from a revision across the areas described below, including the use of more precise language, clarifying that association does not equal causation, and being specific with practical implications of the study for clinicians, pharmacists, and regulatory authorities.

2. Title and Abstract

(2.I). In spontaneous reports, an association of adverse events with a particular drug use does not necessarily equal causation. Thus, for clarity in the title, “adverse events of SZC” might be better phrased as “adverse events reported for SZC” or “adverse events associated with SZC.”

(2.II). The objective section of the abstract can be more precise by restating "investigate the occurrence" to “investigate frequencies, types, and signals of ADEs associated with SZC….”

(2.III). Also, in the objective section of the abstract, the phrase “provide a reference” is vague. This can be rephrased as “to inform clinicians of possible safety concerns linked with SZC in real-life usage.”

(2.III). The methods section of the abstract did not mention the specific time range that was investigated from the database. This is important to guide reproducibility.

(2.IV). The result section of the abstract should specify the particular PTs that made the positive signal cut off.

(2.V). In the conclusion part, the phrase “common ADEs of SZC were confirmed” seems overstated and could be softened to “common ADEs reported for SZC” or “common ADEs associated with SZC.”

3. Introduction

(3.I). “Etc” is informal and should be removed from the sentence “This condition is a life-threatening disease associated with severe cardiac arrhythmias, sudden death, etc.” Be specific.

(3.II). Provide citation(s) for the claim on the frequency of hyperkalemia among patients with CKD, diabetes,…

(3.III). Starting two successive sentences the same way, “And SZC was observed to” “And it is not easy to induce” is redundant. Consider starting the second sentence with a different opening line, like “Additionally,” or “Further,”

4. Methods

(4.I). This section is silent on the study’s strategy for data deduplication or cleaning, which is important for processing data mined from FAERS. If this were done, it should be mentioned.

(4.II.) It is not clear if the primary suspect drugs were the ones considered or not, which will help inform the signal strength. This needs to be clarified, with a justification for the choice made.

(4.III). It is not clear if PRR and ROR were applied to both FAERS and VigiAccess or just one of them. This clarification is important.

(4.IV). There should be a mention of the strategies the authors used for handling the limitations of the data sources, such as underreporting, duplicate entries, or reporting bias.

(4.V). It is not clear which software(s) was used for the data analysis.

5. Results

(5.I). In section 3.1, the phrase “the main reporting continents is America” can be rephrased to “Reports primarily originated from North America.”

(5.II). In section 3.2, the comparison of SOC-level distributions across both databases and the identification of consistency between FAERS and VigiAccess is great. However, reporting of raw frequencies without normalization may be misleading, especially if the two databases differ in size.

(5.III). In section 3.3, it should be clarified whether the deaths talked about are adjudicated or unconfirmed.

(5.IV). There could be an additional subsection with a figure showing trends of ADEs across the years within the timeline being investigated.

6. Discussion

(6.I). The phrase “we analyzed the ADEs of SZC” could be rephrased to “we analyzed the ADEs associated with SZC.”

(6.II). The phrase “This study found that there the number of male patients…” should be rewritten as “This study found that the number of male patients…”

(6.IV). Some limitations were mentioned. However, there should be additional mentions of other renowned limitations of spontaneous ADEs reporting, like underreporting, poor documentation, difficulty in establishing causality, and the potential for false-positive reports.

7. Conclusion

(7.I). The conclusion does not offer specific ways the findings can help inform clinicians. How about some specific recommendations, like the need to monitor sodium levels and screen for cardiac risks before administration of SZC?

8. Final Recommendations

(8.I). The manuscript seeks to fill a critical knowledge gap on a clinically important topic, but will benefit from a revision across the areas described above.

Reviewer #2: Title: Real-World Study of Adverse Events of Sodium Zirconium Cyclosilicate Based on US FDA Adverse Event Reporting System and VigiAccess Database

I would like to thank the authors for submitting their work titled "Real-World Study of Adverse Events of Sodium Zirconium Cyclosilicate Based on US FDA Adverse Event Reporting System and VigiAccess Database". I would also like to thank the editors for giving me the opportunity to review and comment on this submission.

In their contribution, the authors address an interesting and contemporary area of research: investigating the occurrence of adverse drug events (ADEs) associated with the use of sodium zirconium cyclosilicate for treating hyperkalemia. The aim is to provide a reference for safe clinical practice.

However, several aspects require clarification and improvement.

#Introduction and Discussion

Novelty and added value:

The manuscript does not sufficiently clarify the differences and added value compared to existing studies, particularly the recently published analysis by Liu et al. (2024, PLOS ONE: https://journals.plos.org/plosone/article?id=10.1371/journal.pone.0320585). A clear discussion on how this work differs from previous research in terms of methodology and substance is needed.

Language and style:

The manuscript requires careful editing to improve clarity and ensure a consistent scientific tone. For example, consider the following sentence: “The adsorption capacity of sodium zirconium cyclosilicate for K+ was found to be significantly greater than that of other ions. And sodium zirconium cyclosilicate was observed to have no impact on the utilisation of RAASis drugs. And it is not easy to induce electrolyte disorders in the body, and has no significant effect on liver function, nutritional status and other indicators, showing good tolerability and safety.”

#Methods

Methodology details:

Although the data sources (FAERS and VigiAccess), the retrieval period and the use of OpenVigil 2.1 for data extraction are clearly stated, the description of the disproportionality analysis (ROR and PRR) is insufficient. The manuscript should provide more detailed information on how the ROR and PRR were calculated and specify the software or tools used for the statistical analysis.

Data presentation and consistency:

There are inconsistencies in Table 1 between the absolute numbers and their respective percentages. For example, 380 females do not represent 24.46% of 1384. Please carefully review and correct these discrepancies.

Reference population in tables:

In Tables 2 (SOC distribution of ADEs) and 3 (Top 30 PTs by number of reported cases), the reference population or denominator for the presented values remains unclear. A clearer explanation of the basis for these proportions is necessary to ensure interpretability.

Statistical analysis and group comparisons:

The analysis should go beyond simple descriptive observations such as 'there were more males than females'. Appropriate statistical tests should be conducted to determine whether any observed differences between groups, including but not limited to gender, are statistically significant.

Figure quality:

Figure 1 is displayed at too small a size, which limits the reader’s ability to discern important details. A larger, higher-resolution version would improve clarity.

Subgroup analyses:

Including subgroup analyses, particularly with regard to gender differences, would add significant value to the findings and should be considered.

It is not apparent from the manuscript whether the study adheres to the relevant reporting guidelines.

Addressing these points would significantly enhance the scientific rigour, clarity, and overall impact of the manuscript, but substantial revisions are needed before it can be considered suitable for publication.

6. PLOS authors have the option to publish the peer review history of their article (what does this mean?). If published, this will include your full peer review and any attached files.

Reviewer #1: **Yes: **Charles E. Ugwu

Reviewer #2: No

---

## [Author Response · Author response to Decision Letter 1]

10 Jul 2025

Response Letter

Manuscript Number: PONE-D-25-21593

Title: Real-World Study of Adverse Events of Sodium Zirconium Cyclosilicate Based on US FDA Adverse Event Reporting System and VigiAccess Database

Dear Editors and Reviewers,

We sincerely appreciate the opportunity to revise our manuscript titled “Real-World Study of Adverse Events of Sodium Zirconium Cyclosilicate Based on US FDA Adverse Event Reporting System and VigiAccess Database” (Manuscript ID: PONE-D-25-21593). We are grateful to the reviewers for their constructive and insightful comments, which have helped us significantly improve the quality of our work.

We have carefully addressed all the reviewers’ comments and made substantial revisions to the manuscript accordingly. Below, we provide a point-by-point response to each comment, detailing the changes implemented in the revised version. All modifications in the manuscript are highlighted in yellow for easy reference.

Response to Dr. Ayipo (Academic Editor):

Thank you very much for your time and for giving us the opportunity to revise our manuscript. We have carefully addressed all the points raised in the editorial comments and reviewer suggestions, as well as the journal’s requirements. The revised manuscript has now been uploaded accordingly. Should there be any further questions or additional revisions needed, please do not hesitate to let me know. Thank you again for your consideration and support.

Response to Reviewer 1:

First and foremost, we sincerely appreciate Dr. Ugwu (Reviewer 1) for his detailed and comprehensive comments. Below, we provide point-by-point responses to each of the suggestions.

1.Comment: Title and Abstract (2.I). In spontaneous reports, an association of adverse events with a particular drug use does not necessarily equal causation. Thus, for clarity in the title, “adverse events of SZC” might be better phrased as “adverse events reported for SZC” or “adverse events associated with SZC.”

Response: “adverse events of SZC” has been changed to “adverse events associated with SZC” in the title.

2.Comment: Title and Abstract (2.II). The objective section of the abstract can be more precise by restating “investigate the occurrence” to “investigate frequencies, types, and signals of ADEs associated with SZC….”

Response: “investigate the occurrence” has been changed to “investigate frequencies, types, and signals of ADEs associated with SZC”in the objective section of the abstract.

3.Comment: Title and Abstract (2.III). In the objective section of the abstract, the phrase “provide a reference” is vague. This can be rephrased as “to inform clinicians of possible safety concerns linked with SZC in real-life usage”. The methods section of the abstract did not mention the specific time range that was investigated from the database.

Response: “provide a reference” has been changed to “to inform clinicians of possible safety concerns linked with SZC in real-life usage.” in the objective section of the abstract. The specific retrieval time of the database has been supplemented in the methods section of the abstract, which is expressed as “Data extraction from FAERS was performed using OpenVigil 2.1, covering reports from the first quarter of 2004 through the third quarter of 2024. The VigiAccess database was retrieved for reports up to February 5, 2025.”

4.Comment: Title and Abstract (2.IV). The result section of the abstract should specify the particular PTs that made the positive signal cut off.

Response: As recommended, we have revised the results section of the abstract to explicitly list the key PTs that met the positive signal criteria. Please see the highlighted changes in the abstract.

5.Comment: Title and Abstract (2.V). In the conclusion part, the phrase “common ADEs of SZC were confirmed” seems overstated and could be softened to “common ADEs reported for SZC” or “common ADEs associated with SZC.”

Response: “common ADEs of SZC were confirmed” has been revised to “common ADEs associated with SZC.”

6.Comment: Introduction (3.I). “Etc” is informal and should be removed from the sentence “This condition is a life-threatening disease associated with severe cardiac arrhythmias, sudden death, etc.”

Response: “Etc” has been removed. The sentence has been revised to: “This disease can lead to life-threatening conditions such as severe cardiac arrhythmias and sudden death.”

7. Comment: Introduction (3.II). Provide citation(s) for the claim on the frequency of hyperkalemia among patients with CKD, diabetes,…

Response: Citation(s) has been supplemented.

8.Comment: Introduction (3.III). Starting two successive sentences the same way, “And SZC was observed to” “And it is not easy to induce” is redundant. Consider starting the second sentence with a different opening line, like “Additionally,” or “Further,”

Response: Thanks for this linguistic improvement suggestion. The second sentence has been amended to start with “Additionally,.”

9.Comment: Methods (4.I). This section is silent on the study’s strategy for data deduplication or cleaning, which is important for processing data mined from FAERS.

Response: Thanks for this comment. In this study, OpenVigil 2.1 was used to extract and analyse data from the FAERS database, OpenVigil 2.1 itself has already done the data cleaning and deduplication, which has been mentioned in the discussion section. And more references have been added in the methods section.

10.Comment: Methods (4.II.) It is not clear if the primary suspect drugs were the ones considered or not, which will help inform the signal strength. This needs to be clarified, with a justification for the choice made.

Response: Only AE reports with SZC as the primary suspect drug were extracted and analysed for signal intensity in FAERS in this study. It is mentioned in the methods section.

11.Comment: Methods (4.III). It is not clear if PRR and ROR were applied to both FAERS and VigiAccess or just one of them.

Response: Thanks for raising this important point. PRR and ROR only apply to FAERS. VigiAccess does not support the calculation of PRR/ROR due to limited data granularity (e.g., complete background population data is not available). This point has now been clarified in the methods section.

12.Comment: Methods (4.IV). There should be a mention of the strategies the authors used for handling the limitations of the data sources, such as underreporting, duplicate entries, or reporting bias.

Response: As mentioned in the response #9, the data in FAERS extracted by OpenVigil is deduplicated and cleaned. Underreporting and reporting bias are inherent limitations of spontaneous reporting database, which have been mentioned in the discussion section (limitations paragraph).

13.Comment: Methods (4.V). It is not clear which software(s) was used for the data analysis.

Response: Software for data analysis has been supplemented in the methods section.

14.Comment: Results (5.I). In section 3.1, the phrase “the main reporting continents is America” can be rephrased to “Reports primarily originated from North America.”

Response: It has been modified as suggested.

15.Comment: Results (5.II). In section 3.2, the comparison of SOC-level distributions across both databases and the identification of consistency between FAERS and VigiAccess is great. However, reporting of raw frequencies without normalization may be misleading, especially if the two databases differ in size.

Response: We quite agree that the data of two databases should not be directly compared. The SOC-level analysis in Section 3.2 was only used to analyse the SOC distribution of AEs reported by the two databases, not for cross-database comparison. Summarizing the data in one table may lead to some misleading, and it has been divided into two tables.

16.Comment: Results (5.III). In section 3.3, it should be clarified whether the deaths talked about are adjudicated or unconfirmed.

Response: The “death” mentioned is an outcome of patients who have used SZC, does not mean that it was directly caused by SZC. This AE has been discussed further in the discussion section.

17.Comment: Results (5.IV). There could be an additional subsection with a figure showing trends of ADEs across the years within the timeline being investigated.

Response: Yearly trends in the number of AE reports have been presented in results section 3.1 (Reporting year).

18.Comment: Discussion (6.I). The phrase “we analyzed the ADEs of SZC” could be rephrased to “we analyzed the ADEs associated with SZC.”

Response: Revised according to the comment.

19.Comment: Discussion (6.II). The phrase “This study found that there the number of male patients…” should be rewritten as “This study found that the number of male patients…”

Response: Revised according to the comment.

20.Comment: Discussion (6.III). Some limitations were mentioned. However, there should be additional mentions of other renowned limitations of spontaneous ADEs reporting, like underreporting, poor documentation, difficulty in establishing causality, and the potential for false-positive reports.

Response: The limitation section has been revised in accordance with the comments.

21.Comment: Conclusion (7.I). The conclusion does not offer specific ways the findings can help inform clinicians. How about some specific recommendations, like the need to monitor sodium levels and screen for cardiac risks before administration of SZC?

Response: We sincerely appreciate this constructive suggestion. As recommended, we have now added specific clinical recommendations to the conclusion section: “In clinical use, healthcare professionals need to pay special attention to monitoring patients' blood sodium levels, preventing cardiac risks such as chronic heart failure and gastrointestinal system risks such as ileus and intestinal perforation, as well as other risk signals detected.”

Response to Reviewer 2:

We sincerely appreciate Reviewer 2’s constructive comments and valuable suggestions. Below we provide point-by-point responses to all the raised issues.

1.Comment: The manuscript does not sufficiently clarify the differences and added value compared to existing studies, particularly the recently published analysis by Liu et al. (2024, PLOS ONE: https://journals.plos.org/plosone/article?id=10.1371/journal.pone.0320585). A clear discussion on how this work differs from previous research in terms of methodology and substance is needed.

Response: Thanks for the suggestion. We have carefully studied the work mentioned above, and have revised the Discussion section to clarify the novelty and distinctions between our work and Liu et al. (2024).

Key points include: 1. In our study, the data of two databases are extracted, and the data of WHO-VigiAccess are analyzed besides FAERS. The data sources and geographical coverage of the two databases are different, which can provide a more comprehensive perspective for the study of AEs. 2. We used OpenVigil 2.1 to mine and analyse the data from FAERS. OpenVigil 2.1 cleansed the data based on demographic file and drug file, and loaded only the reports with complete drug name information, which sacrificed the original sample size to a certain extent but ensured a higher quality of data. 3. Compared with the published research mentioned above, our study added data from 2024. SZC has been on the market for a very short time and the number of reported AEs is relatively small, but the number of reports has increased significantly in the last two years. In our research, the number of reports in 2024 accounted for 28.03% in FAERS and 39.79% in VigiAccess. 4. In terms of research results, the SOCs and PTs involved in the positive signals are different, and the results of gender differences are also different, which may be due to the sample size as well as different data extraction methods.

2.Comment: The manuscript requires careful editing to improve clarity and ensure a consistent scientific tone. For example, consider the following sentence: “The adsorption capacity of…showing good tolerability and safety.”

Response: Revised according to the comment. Other similar issues have also been revised and highlighted in yellow.

3.Comment: The manuscript should provide more detailed information on how the ROR and PRR were calculated and specify the software or tools used for the statistical analysis.

Response: Detailed information about how to calculate ROR and PRR, and the software used has been supplemented in the Method section.

4.Comment: There are inconsistencies in Table 1 between the absolute numbers and their respective percentages. For example, 380 females do not represent 24.46% of 1384.

Response: Sorry for our careless mistake. “24.46” has been revised to “27.46”, and other data in the table have been rechecked.

5.Comment: In Tables 2 (SOC distribution of ADEs) and 3 (Top 30 PTs by number of reported cases), the reference population or denominator for the presented values remains unclear. A clearer explanation of the basis for these proportions is necessary to ensure interpretability.

Response: Appreciate the valuable comment. Denominators of the values listed have been supplemented and highlighted in yellow in the paragraphs.

6.Comment: The analysis should go beyond simple descriptive observations such as 'there were more males than females'. Appropriate statistical tests should be conducted.

Response: Already revised. Section 3.1 of the results has been revised as "There were significantly more males than females in both databases (FAERS: 48.19% vs 27.46%, p < 0.001; VigiAccess: 52.50% vs 30.83%, p < 0.001)"

7.Comment: Figure 1 is displayed at too small a size, which limits the reader’s ability to discern important details. A larger, higher-resolution version would improve clarity.

Response: The figures have been revised as suggested, with high-resolution files uploaded separately in compliance with the journal's guidelines.

8.Comment: Including subgroup analyses, particularly with regard to gender differences, would add significant value to the findings and should be considered.

Response: Thanks for this constructive comment. As recommended, we have now conducted subgroup analyses stratified by gender (Results section 3.3; Figs 3 and 4), and further discussed these results in the Discussion section.

---

## [Decision Letter · Decision Letter 1]

18 Sep 2025

Real-world study of adverse events associated with sodium zirconium cyclosilicate based on FDA adverse event reporting system and VigiAccess database

PONE-D-25-21593R1

Dear Dr. Jia,

We’re pleased to inform you that your manuscript has been judged scientifically suitable for publication and will be formally accepted for publication once it meets all outstanding technical requirements.

Kind regards,

Yusuf Oloruntoyin Ayipo, Ph.D

Academic Editor

PLOS ONE

Additional Editor Comments (optional):

The submission is scientifically sound for publication in this title, and all the concerns raised by the respective reviewers regarding the manuscript quality have been satisfactorily addressed. I hereby recommend the manuscript for publication in the current version.

Reviewers' comments:

Reviewer's Responses to Questions

**Comments to the Author**

1. If the authors have adequately addressed your comments raised in a previous round of review and you feel that this manuscript is now acceptable for publication, you may indicate that here to bypass the “Comments to the Author” section, enter your conflict of interest statement in the “Confidential to Editor” section, and submit your "Accept" recommendation.

Reviewer #1: All comments have been addressed

Reviewer #3: (No Response)

2. Is the manuscript technically sound, and do the data support the conclusions?

Reviewer #1: Yes

Reviewer #3: Yes

3. Has the statistical analysis been performed appropriately and rigorously? 

Reviewer #1: Yes

Reviewer #3: Yes

4. Have the authors made all data underlying the findings in their manuscript fully available?

Reviewer #1: Yes

Reviewer #3: Yes

5. Is the manuscript presented in an intelligible fashion and written in standard English?

Reviewer #1: Yes

Reviewer #3: Yes

6. Review Comments to the Author

Reviewer #1: All the issues I raised have been addressed satisfactorily. All the issues I raised have been addressed satisfactorily.

Reviewer #3: This study, which investigates adverse drug events (ADEs) associated with sodium zirconium cyclosilicate (SZC) a treatment for hyperkalemia using data from the FAERS and VigiAccess databases, is timely and valuable. Leveraging large pharmacovigilance datasets such as FAERS and VigiAccess provides important real-world evidence that can inform the clinical use of SZC. The Methods section is generally well-structured and clearly presented, with appropriate database usage and analytical strategies. However, a few clarifications could further enhance the transparency and reproducibility of the study:

Could the authors specify the criteria used for inclusion or exclusion of ADE reports (e.g., completeness of data, primary suspect status)?

The statistical tool(s) or software used for analysis were not mentioned; including this would aid in similar future study.

MedDRA version 27.0 was used appropriately for coding; however, please clarify whether this was the most current version available at the time of analysis.

7. PLOS authors have the option to publish the peer review history of their article (what does this mean?). If published, this will include your full peer review and any attached files.

Reviewer #1: No

Reviewer #3: No

---

## [Editor Report · Acceptance letter]

PONE-D-25-21593R1

PLOS ONE

Dear Dr. Jia,

I'm pleased to inform you that your manuscript has been deemed suitable for publication in PLOS ONE. Congratulations! Your manuscript is now being handed over to our production team.

Kind regards,

on behalf of

Dr. Yusuf Oloruntoyin Ayipo

Academic Editor

PLOS ONE